# The Mediating Effect of Coping Strategies and Emotion Regulation in the Relationship between Impulsivity, Metacognition, and Eating Disorders

**DOI:** 10.3390/nu16121884

**Published:** 2024-06-14

**Authors:** Ana Estévez, Janire Momeñe, Laura Macía, Iciar Iruarrizaga, Leticia Olave, Gema Aonso-Diego

**Affiliations:** 1Department of Psychology, Faculty of Health Sciences, University of Deusto, Avda. de las Universidades 24, 48007 Bilbao, Spain; janiremomene@deusto.es (J.M.); laura.macia@deusto.es (L.M.); gema.aonso@deusto.es (G.A.-D.); 2Department of Experimental Psychology, Cognitive Psychology and Speech & Language Therapy, Faculty of Psychology, Complutense University of Madrid, 28040 Madrid, Spain; iciariru@psi.ucm.es (I.I.); leticiaolave@ucm.es (L.O.)

**Keywords:** eating disorders, gender differences, impulsivity, coping strategies, metacognition

## Abstract

Background. Risk factors for eating disorders are multifaceted and complex, so it is crucial to elucidate the role of executive functions, including impulsivity and metacognition, and coping strategies in the severity of eating behaviors. The study aims were threefold: (1) to analyze gender differences in impulsivity, metacognition, coping strategies, emotion regulation, and eating disorders; (2) to examine the correlation between the study variables; and (3) to test the mediating role of coping and emotion-regulation strategies in the relationship between metacognition, impulsivity, and eating disorders. Methods. A total of 1076 participants (*M*_age_ = 21.78, *SD* = 5.10; 77.7% women) completed a set of questionnaires. Two mediation analyses were conducted to test the mediating role of coping strategies, including emotion regulation, in the relationship between executive functions (i.e., impulsivity and metacognition) and eating disorders. Results. Women displayed higher coping strategies, specifically emotional expression, wishful thinking, and social support, whereas men presented greater social withdrawal. Mediational analyses showed a significant association between impulsivity, metacognition, and eating disorders, whose relationship was partially mediated by coping strategies and mainly by emotion regulation. Conclusion. Interventions based on coping strategies and emotion regulation could be a feasible and effective option to deal with eating disorders among the young population.

## 1. Introduction

Eating disorders constitute a spectrum of psychiatric conditions characterized by disturbances in eating behaviors, attitudes toward food, body weight, and shape. These disorders significantly impact individuals’ physical and mental health (e.g., depression, anxiety) [1], including suicide [2], and notably, the mortality rate for eating disorders is strikingly high [3,4,5]. Among the most prevalent eating disorders are anorexia nervosa, bulimia nervosa, and binge eating disorder, each presenting distinct clinical features and diagnostic criteria [6].

The etiology of eating disorders is multifaceted, influenced by a complex interplay of genetic, biological [7], psychological (e.g., stressful life events) [8,9], and sociocultural factors (e.g., cultural norms) [10]. Two variables that have received the most attention are those related to executive function, specifically lack of control or impulsivity, and metacognitive strategies.

Impulsivity, broadly defined as a predisposition towards rapid, unplanned reactions to internal or external stimuli without consideration of potential consequences, has been deeply studied within the realm of eating disorders, showing a different profile of impulsivity for each diagnosis [11,12]. Previous evidence suggests a strong relationship between impulsivity and different eating disorders (i.e., anorexia nervosa, bulimia nervosa, and binge eating disorder) [13,14,15,16,17], as well as the risk of developing an eating disorder [18]. Impulsivity manifests in binge eating episodes, purging behaviors, or excessive exercise, which are undertaken without forethought or control [19,20].

On the other hand, metacognition, the process of thinking about one’s own thinking, beliefs, and cognitive processes, has emerged as a critical factor in understanding eating behaviors and the development and maintenance of eating disorders. This variable has been related to a large number of psychological disorders [21,22], addictive behaviors [23,24], as well as eating disorders and problematic eating behaviors [25,26,27,28,29], both in anorexia and bulimia [30,31,32] and binge eating [33].

Despite the large number of studies confirming the association between impulsivity and metacognitive strategies with eating disorders, the causal mechanism remains unclear. Some studies have pointed to the importance of coping strategies, including emotion regulation, in explaining this relationship [34,35,36,37].

Coping and emotion regulation strategies have been widely studied in relation to eating disorders [38,39,40,41,42]. Nonetheless, to our knowledge, no previous studies have analyzed the mediating role of coping strategies and emotion regulation in the relationship between metacognition or impulsivity and eating disorders. However, it has been studied in other psychological variables, such as anxiety [43,44], attachment [45], excessive worry [46], posttraumatic symptoms [47], Internet addiction [48], and overall mental health [49], which show elevate comorbidity with eating disorders [1,50].

Based on the Self-Regulatory Executive Function model (S-REF) [51,52,53,54], metacognitive beliefs activate a combination of coping strategies. In other words, whereas metacognition refers to the awareness and understanding of one’s own cognitive processes, including thoughts, beliefs, and strategies, coping strategies encompass the behavioral, cognitive, and emotional efforts employed to manage stressors and adapt to adverse circumstances. Positive metacognitive beliefs (e.g., self-efficacy and perceived control over stressors) promote the utilization of proactive coping strategies (e.g., problem-solving, seeking social support, or positive reappraisal). Conversely, negative metacognitive beliefs, such as catastrophic thinking, may predispose individuals to employ maladaptive coping strategies (e.g., avoidance, rumination, social withdrawal), which exacerbate emotional distress and increase vulnerability to engaging in binge eating, purging, or restrictive eating behaviors, as well as maintaining eating disorders over time.

In this line, impulsivity often undermines the effectiveness of coping efforts by leading individuals to employ maladaptive or reactive strategies in response to stressors [55,56,57,58]. Impulsive individuals may resort to impulsive coping mechanisms, such as avoidance or dysfunctional emotion regulation, as a solution to alleviate distress in the short term.

Despite the amount of research on this issue, no study has tested the mediating role of coping strategies and emotion regulation in the relationship between metacognition or impulsivity and eating disorders. Furthermore, previous studies focusing on these variables (i.e., metacognition, impulsivity, coping strategies, and emotion regulation) were conducted with a clinical population [14,25,30,33], and mostly among the adolescent population [26,27,59]. Also, to our knowledge, no previous study has been conducted among the Spanish population in which eating-related characteristics could differ from other countries [60]. Understanding the role of metacognitive beliefs and impulsivity in shaping eating behaviors has important implications for the development of targeted interventions for eating disorders.

Given this background, the study aims were threefold: (1) to analyze gender differences in the study variables (i.e., eating disorders, impulsivity, metacognition, emotion regulation, and some coping strategies); (2) to examine the correlation between the study variables; and (3) to test the mediating role of coping and emotion-regulation strategies in the relationship between metacognition, impulsivity, and eating disorders.

## 2. Material and Methods

### 2.1. Participants and Procedure

The study sample included 1076 participants (*M*_age_ = 21.78, *SD* = 5.10, aged between 18 and 61 years). Most participants were women (77.7%, *n* = 836), 21.7% (*n* = 233) were men, and the remaining seven participants (0.70%) were trans men. The majority were students (78.6%, *n* = 845) and single (93.1%, *n* = 1000). Finally, 34.57% (*n* = 372) of the participants had received psychological treatment at some point in their lives, of which 5.10% (*n* = 19) were for eating disorders.

The recruitment was conducted through two channels: online and face-to-face. For online recruitment, surveys were conducted through a SurveyMonkey platform. For the face-to-face channel, participants were recruited at the Complutense University of Madrid. Participants filled out the questionnaires in the classroom during teaching hours through pencil and paper or an online survey in the university classrooms. The only exclusion criterion was being under 18 years of age.

All participants provided written informed consent before study participation, indicating that they had read the study information and agreed to participate voluntarily. The study adhered to the ethical principles of the Declaration of Helsinki and was approved by the university ethics committee (ref. ETK-38/23-24). Participants were assured of the confidentiality and anonymity of their responses, as well as their voluntary participation. The participants did not receive any kind of compensation.

### 2.2. Instruments

All participants completed a battery in which sociodemographic data (i.e., age, gender, working status, and marital status) were collected. Additionally, participants completed four questionnaires concerning eating disorders, metacognition, impulsivity, coping strategies, and emotion regulation.

#### 2.2.1. Eating Disorders

The Eating Disorder Inventory (EDI-2) [61], validated in Spain by Muro-Sans et al. (2006) [62], is a widely used self-report questionnaire designed to assess various aspects of disordered eating behaviors and attitudes. This inventory included a total of 91 items divided into 11 scales that evaluate central aspects of eating disorders, that is, drive for thinness, bulimia, body dissatisfaction, ineffectiveness, perfectionism, interpersonal distrust, interoceptive awareness, maturity fears, asceticism, impulse regulation, and social insecurity. All items are rated using a Likert-type scale, where individuals rate the extent to which each statement applies to them, ranging from 0 (“never”) to 5 (“always”). The internal consistency measured by Cronbach’s α of the total scale was 0.95.

#### 2.2.2. Metacognitive Strategies

Metacognitive strategies were assessed with the homonymous subscale of the Inventory of Cognitive Activity in Anxiety Disorders, subscale for obsessive-compulsive disorder (*Inventario de Actividad Cognitiva en Trastornos de Ansiedad—Trastorno Obsesivo-Compulsivo*; IACTA-TOC) [63]. The total inventory consists of 21 items, including the metacognition subscale, which comprises 7 items rated on a Likert-type scale (ranging from 0 “almost never” to 4 “almost always”). The subscale assessed the frequency of certain thoughts in individuals (e.g., “I place great importance on the thoughts that occupy my mind”, “at times, I may mistake my thoughts for reality”, “I am vigilant and attentive to situations related to my thoughts”). In our sample, Cronbach’s alpha was 0.88 for the metacognition subscale.

#### 2.2.3. Impulsivity

The Barratt Impulsiveness Scale (BIS-11) [64], adapted to Spanish by Oquendo et al. (2001) [65], is a self-report questionnaire assessing several aspects of impulsivity. The BIS-11 is a 30-item questionnaire, rated on a 4-point Likert-type scale, ranging from 1 (never/rarely) to 4 (almost always/always). Items are grouped into three subscales: attentional impulsiveness, motor impulsiveness, and non-planning impulsiveness. In the present study, Cronbach’s alpha was 0.76 for the global scale.

#### 2.2.4. Coping Strategies

The Coping Strategies Inventory (CSI) [66], validated in Spanish by Cano-García et al. (2007) [67], is a self-report measure designed to assess coping strategies employed by individuals to manage stress and adversity. The CSI includes 40 items grouped into eight factors: problem-solving, cognitive restructuring, social support, emotional expression, problem avoidance, wishful thinking, social withdrawal, and self-criticism. Among the eight factors, there are four with positive valence, that is, higher scores indicate adaptive coping strategies (i.e., problem-solving, emotional expression, social support, and cognitive restructuring), whereas the four remaining factors have a negative valence, namely, higher scores indicate maladaptive coping strategies (i.e., self-criticism, wishful thinking, problem avoidance, and social withdrawal). All items are rated on a five-point Likert-type scale ranging from 0 (not at all) to 4 (completely). The internal consistency in the present study ranged from 0.69 in the social withdrawal subscale to 0.88 in the self-criticism subscale.

#### 2.2.5. Emotion Regulation

The Difficulties in Emotion Regulation Scale (DERS) [68], adapted to Spanish by Hervás and Jódar (2008) [69], was used to assess difficulties in the awareness, understanding, or modulation of emotion. The Spanish validation comprises a total of 28 items rated on a Likert-type scale (1 = almost never, 5 = almost always) assessing five emotion-regulation deficits: (1) non-acceptance of emotions; (2) daily interference (‘difficulties engaging in goal-directed behavior’ in the original subscale); (3) emotional inattention (‘lack of emotional awareness’ in the original subscale); (4) emotional confusion (‘lack of emotional clarity’ in the original subscale); and (5) lack of emotional control (‘limited access to emotion regulation strategies’ and ‘impulse control difficulties’ in the original subscales). In the present study, the scale has shown adequate psychometric properties (α = 0.93).

### 2.3. Data Analysis

Descriptive statistics and frequencies were computed to summarize the sociodemographic characteristics and the study variables. Independent samples *t*-tests were conducted to examine gender differences in the study variables. Effect sizes were calculated using Cohen’s *d* [70]. Pearson correlation coefficients were computed to assess the relationships between study variables.

Two mediation analyses were performed to test the mediating role of emotion regulation and coping strategies in the relationship between impulsivity and metacognition and eating disorders. Therefore, we considered coping strategies (i.e., problem-solving, cognitive restructuring, social support, emotional expression, problem avoidance, wishful thinking, social withdrawal, and self-criticism) and emotion regulation as mediation variables. Given the high correlation between the subscales of emotion regulation and the total score (ranging between 0.723 and 0.878) and between the subscale of impulsivity and the total score (ranging between 0.728 and 0.788), we deemed the total score in these questionnaires.

This analysis was conducted with Process 4 for SPSS Statistics 28 [71], using 5000 bootstrapping samples. All analyses were performed using the SPSS package, and the confidence levels were set at 95%.

## 3. Results

The study found significant gender differences in age, body mass index (BMI), weight, and height (all *p*-values ≤ 0.001), with women being younger and having lower BMI, weight, and height. Additionally, more men were married and employed. Conversely, there were no gender differences in the study variables, including eating disorders, impulsivity, metacognition, emotion regulation, and some coping strategies. However, there were gender differences in emotional expression, wishful thinking, social support, and social withdrawal, where women scored higher (see Table 1).

Bivariate correlation analyses revealed a strong relationship among the study variables. The only non-significant relationships found were between specific coping strategies and eating disorders (*r* = 0.06), impulsivity (*r* ≤ −0.055), metacognition (*r* ≤ 0.042), and emotion regulation (*r* = 0.059) (see Table 2).

The mediation analyses showed that impulsivity and metacognition have significant indirect effects on eating disorders through the mediator variables (i.e., coping strategies and emotion regulation). Both mediation models achieved similar results with a significant direct effect of impulsivity and metacognition on eating disorders, emotion regulation, and coping strategies (except for cognitive restructuring and problem avoidance). Additionally, all indirect effects through emotion regulation and coping strategies were statistically significant, except for the pathway mediated by problem-solving, emotional expression, cognitive restructuring, and problem avoidance. Overall, these results demonstrated the intricate connections between impulsivity, metacognition, and eating disorders, whose relationship was partially mediated by emotion regulation and coping strategies, as evidenced by the statistically significant direct and total effects (see Table 3 and Table 4). The two mediating models are displayed in Figure 1 and Figure 2, respectively.

## 4. Discussion

The current study aimed to test the mediating role of coping strategies, including emotion regulation, in the relationship between impulsivity and metacognition, and eating disorders. Two results are underlined: (1) Women displayed higher coping strategies, specifically emotional expression, wishful thinking, and social support, whereas men presented greater social withdrawal; and (2) there was a significant association between impulsivity, metacognition, and eating disorders, whose relationship was partially mediated by coping strategies and mainly by emotion regulation.

The results did not find gender differences in eating disorder severity, impulsivity levels, metacognition, or emotion regulation, although differences between men and women were found in some coping strategies. In contrast to the present findings, previous studies have consistently indicated that women exhibit a higher prevalence of eating disorders and greater severity, both in the general population [72,73] and in young adults [74]; higher emotion regulation [75]; and higher scores on some coping strategies [76], although other studies suggest that men score higher on other coping strategies (e.g., avoidance, distraction, and social diversion) [77].

On the other hand, there was a high correlation between coping strategies, especially emotion regulation, and the severity of eating disorders. Emotion regulation has been widely studied as a transdiagnostic construct and also its relationship with several psychological disorders [78,79,80,81], including eating disorders [82,83,84,85]. Likewise, a strong relationship was also observed between executive functions, specifically impulsivity and metacognition, with eating disorders, coping strategies, and emotion regulation. Theories such as that proposed by Sharp et al. (2013) [86] indicate that women with eating disorders may articulate their mental states extensively and in detail, demonstrating a cognitive understanding of affectivity but with minimal connection to experiential or affective core [87]. In this regard, cognitive strategies characterized by excessive rumination, for instance, tend to focus on issues such as weight, food, and body image, potentially serving as a cognitive-emotional avoidance strategy, thus perpetuating the maintenance of the eating problem [88,89,90,91], as they impede cognitive and emotional insight [92].

Finally, the results regarding the mediation analysis yielded similar outcomes for impulsivity and metacognition as independent variables. In both models, coping strategies and emotion regulation partially mediated the relationship between impulsivity and metacognition and eating disorders, except for the pathway mediated by problem-solving, emotional expression, cognitive restructuring, and problem avoidance. This relationship has been demonstrated in other psychological disorders (e.g., anxiety, addictive disorder, mental health) [43,44,48,49], but, to our knowledge, it had not been confirmed in relation to eating disorders until now.

This finding leads to significant clinical implications for the young adult population, as interventions targeting effective coping strategies for dealing with emotional distress may prevent youth from disordered eating. First, it is worth mentioning that therapists’ vulnerabilities, biases, and difficulties must be taken into account in addressing eating disorders [93]. Also, several studies have shown the effectiveness of treatments focused on emotion regulation, metacognition, and coping strategies. The systematic review by Sloan et al. (2017) [94] found emotion regulation to be an effective transdiagnostic intervention for eating disorders and other psychological disorders (e.g., anxiety, depression, personality disorders, substance use). Balzan et al. (2023) [95] showed promising results of a metacognition-based intervention in adolescents with anorexia nervosa. Other studies have also examined the effect of coping-based intervention on mental health with excellent results [96,97], although to our knowledge, no studies have conducted coping-based intervention targeted at eating disorders.

When drawing conclusions from the study, it is important to consider certain limitations. Firstly, this is a cross-sectional study, thus causal relationships between variables cannot be established. Moreover, the sample is predominantly composed of women. Hence, future studies should strive to balance gender representation. Although the age range is broad, the sample predominantly comprises young student populations, thus limiting the generalizability of the results to a broader population. Additionally, data collection relied on self-reports, potentially introducing biases such as social desirability. Lastly, given that the sample consisted of the general population rather than clinical populations, caution must be exercised in extrapolating the findings to individuals with eating disorders.

## 5. Conclusions

In conclusion, understanding eating disorders is paramount for early detection, intervention, and successful treatment outcomes. These findings showed that coping and emotion regulation strategies partially mediated the relationship between executive functions (i.e., impulsivity and metacognition) and eating disorders. Therefore, by elucidating the intricate interplay of executive functions and coping strategies contributing to these disorders, clinicians and researchers can advance our knowledge and develop therapeutic strategies to reduce the impact of eating disorders on individuals and society.

## Figures and Tables

**Figure 1 nutrients-16-01884-f001:**
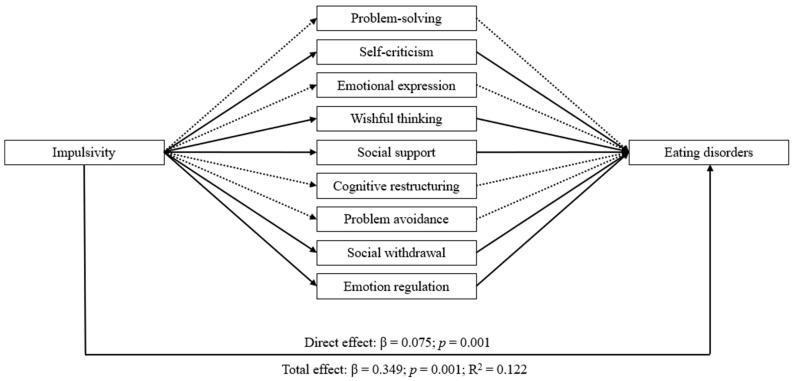
The mediating role of coping strategies and emotion regulation in the relationship between impulsivity and eating disorders. Note: Solid lines indicate significant indirect effects at a 95% confidence interval. Dashed lines indicate non-significant indirect effects at a 95% confidence interval.

**Figure 2 nutrients-16-01884-f002:**
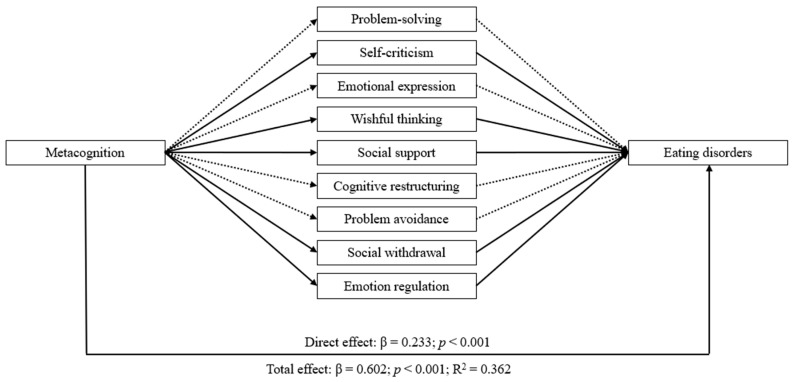
The mediating role of coping strategies and emotion regulation in the relationship between metacognition and eating disorders. Note: Solid lines indicate significant indirect effects at a 95% confidence interval. Dashed lines indicate non-significant indirect effects at a 95% confidence interval.

**Table 1 nutrients-16-01884-t001:** Descriptive statistics of the sociodemographic characteristics and the study variables.

	Total Sample (*N* = 1076)	Male (*n* = 233)	Female (*n* = 834)	*p*-Value	Effect Size
	M (SD)	M (SD)	M (SD)		
Age	21.78 (5.10)	25.19 (6.91)	20.84 (4.00)	<0.001	0.768
Weight (kg)	61.98 (11.78)	73.24 (10.50)	58.84 (10.12)	<0.001	1.396
High (m)	1.66 (0.09)	1.77 (0.07)	1.63 (0.06)	<0.001	2.145
BMI	22.25 (3.24)	23.25 (2.78)	21.97 (3.32)	<0.001	0.419
BMI categories				<0.001	0.142
Underweight (<18.5) ^a^	89 (8.44%)	5 (2.16%)	84 (10.19%)		
Normal weight (18.5–24.9) ^a^	792 (75.07%)	176 (75.19%)	616 (74.76%)		
Overweight (25–29.9) ^a^	145 (13.74%)	45 (19.48%)	100 (12.14%)		
Obesity (>30) ^a^	29 (2.75%)	5 (2.16%)	24 (2.91%)		
Marital status ^a^				<0.001	0.152
Single	993 (93.06%)	209 (89.70%)	784 (94.00%)		
Married	21 (1.97%)	14 (6.01%)	7 (0.84%)		
Other	53 (4.97%)	10 (4.29%)	43 (5.16%)		
Working status ^a^				<0.001	0.350
Student	839 (78.56%)	120 (51.50%)	719 (86.11%)		
Worker	218 (20.41%)	106 (45.49%)	112 (13.41%)		
Other	11 (1.03%)	7 (3.00%)	4 (0.48%)		
Eating disorder	157.88 (55.36)	154.71 (51.05)	158.65 (56.65)	0.355	0.073
Impulsivity	66.72 (9.41)	67.25 (9.28)	66.52 (9.39)	0.307	0.078
Metacognition	9.61 (6.30)	10.21 (6.26)	9.43 (6.29)	0.092	0.124
Coping strategies					
Problem-solving	12.44 (4.57)	12.61 (4.49)	12.41 (4.58)	0.560	0.044
Self-criticism	7.14 (5.25)	7.59 (5.09)	7.04 (5.29)	0.160	0.106
Emotional expression	10.94 (4.81)	10.00 (4.66)	11.23 (4.80)	0.001	0.260
Wishful thinking	13.03 (5.23)	12.23 (5.31)	13.27 (5.20)	0.007	0.198
Social support	14.04 (5.09)	13.07 (4.92)	14.31 (5.11)	0.001	0.247
Cognitive restructuring	10.83 (4.74)	11.06 (4.90)	10.78 (4.68)	0.414	0.058
Problem avoidance	6.98 (4.10)	7.27 (4.08)	6.89 (4.09)	0.200	0.093
Social withdrawal	5.98 (4.23)	6.48 (4.24)	5.82 (4.21)	0.037	0.156
Emotion regulation	63.16 (19.28)	61.53 (16.97)	63.59 (19.85)	0.120	0.112

Note: ^a^ frequency (percentage). M = mean; SD = standard deviation; BMI = body mass index.

**Table 2 nutrients-16-01884-t002:** Correlations of the study variables.

	1	2	3	4	5	6	7	8	9	10	11
Eating disorder											
2. Impulsivity	0.347 **										
3. Metacognition	0.595 **	0.255 **									
4. Problem-solving	−0.291 **	−0.169 **	−0.167 **								
5. Self-criticism	0.426 **	0.164 **	0.396 **	−0.076 *							
6. Emotional expression	−0.163 **	−0.059	0.028	0.389 **	0.043						
7. Wishful thinking	0.328 **	0.079 *	0.299 **	−0.014	0.378 **	0.129 **					
8. Social support	−0.284 **	−0.091 **	−0.159 **	0.356 **	−0.153 **	0.437 **	0.055				
9. Cognitive restructuring	−0.241 **	−0.055	−0.115 **	0.451 **	−0.025	0.254 **	−0.088 **	0.382 **			
10. Problem avoidance	0.060	0.105 **	0.042	−0.056	0.077 *	−0.066 *	0.103 **	0.017	0.358 **		
11. Social withdrawal	0.436 **	0.136 **	0.337 **	−0.237 **	0.407 **	−0.252 **	0.278 **	−0.394 **	−0.139 **	0.254 **	
12. Emotion regulation	0.714 **	0.393 **	0.595 **	−0.259 **	0.417 **	−0.068 *	0.323 **	−0.197 **	−0.190 **	0.059	0.391 **

Note: * *p* ≤ 0.05; ** *p* ≤ 0.01.

**Table 3 nutrients-16-01884-t003:** Standardized direct and indirect effects of impulsivity on eating disorders.

	β	*p*-Value	LLCI	ULCI
Direct effects				
Impulsivity → Eating disorders	0.075	0.002	0.169	0.714
Impulsivity → Emotion regulation	0.402	<0.001	0.688	0.925
Impulsivity → Problem-solving	−0.163	<0.001	−0.107	−0.047
Impulsivity → Self-criticism	0.169	<0.001	0.058	0.128
Impulsivity → Emotion expression	−0.077	0.020	−0.069	−0.006
Impulsivity → Wishful thinking	0.087	0.008	0.012	0.083
Impulsivity → Social support	−0.109	0.001	−0.093	−0.024
Impulsivity → Cognitive restructuring	−0.067	0.042	−0.064	−0.001
Impulsivity → Social withdrawal	0.131	<0.001	0.030	0.086
Impulsivity → Problem avoidance	0.099	0.003	0.015	0.071
Emotional regulation → Eating disorders	0.535	<0.001	1.405	1.719
Problem-solving → Eating disorders	−0.048	0.075	−1.258	0.06
Self-criticism → Eating disorders	0.111	<0.001	0.636	1.741
Emotional expression → Eating disorders	−0.043	0.100	−1.116	0.098
Wishful thinking → Eating disorders	0.080	0.001	0.333	1.376
Social support → Eating disorders	−0.070	0.011	−1.347	−0.173
Cognitive restructuring → Eating disorders	−0.024	0.419	−0.964	0.401
Social withdrawal → Eating disorders	0.104	<0.001	0.656	2.112
Problem avoidance → Eating disorders	−0.010	0.686	−0.814	0.536
Indirect effects				
Emotion regulation	0.215		1.036	1.499
Problem-solving	0.008		−0.005	0.107
Self-criticism	0.019		0.048	0.185
Emotional expression	0.003		−0.003	0.053
Wishful thinking	0.007		0.007	0.089
Social support	0.008		0.007	0.095
Cognitive restructuring	0.002		−0.013	0.043
Social withdrawal	0.014		0.025	0.152
Problem avoidance	−0.001		−0.039	0.025
Total effects (Impulsivity → mediating variables → Eating disorders)	0.349	0.001	1.691	2.400

Note: LLCI = lower-limit confidence intervals; ULCI = upper-limit confidence interval.

**Table 4 nutrients-16-01884-t004:** Standardized direct and indirect effects of metacognition on eating disorders.

	β	*p*-Value	LLCI	ULCI
Direct effects				
Impulsivity → Eating disorders	0.233	<0.001	1.606	2.539
Metacognition → Emotion regulation	0.596	<0.001	1.679	1.996
Metacognition → Problem-solving	−0.181	<0.001	−0.175	−0.085
Metacognition → Self-criticism	0.410	<0.001	0.292	0.389
Metacognition → Emotion expression	0.015	0.656	−0.037	0.059
Metacognition → Wishful thinking	0.287	<0.001	0.188	0.291
Metacognition → Social support	−0.179	<0.001	−0.195	−0.093
Metacognition → Cognitive restructuring	−0.134	<0.001	−0.147	−0.052
Metacognition → Social withdrawal	0.339	<0.001	0.187	0.268
Metacognition → Problem avoidance	0.019	0.565	−0.030	0.055
Emotional regulation → Eating disorders	0.453	<0.001	1.149	1.464
Problem-solving → Eating disorders	−0.038	0.149	−1.099	0.167
Self-criticism → Eating disorders	0.084	0.001	0.365	1.435
Emotional expression → Eating disorders	−0.072	0.005	−1.44	−0.264
Wishful thinking → Eating disorders	0.066	0.006	0.2	1.206
Social support → Eating disorders	−0.057	0.03	−1.196	−0.062
Cognitive restructuring → Eating disorders	−0.036	0.198	−1.086	0.225
Social withdrawal → Eating disorders	0.071	0.009	0.238	1.632
Problem avoidance → Eating disorders	0.009	0.715	−0.527	0.767
Indirect effects				
Emotion regulation	0.27		2.01	2.799
Problem-solving	0.007		−0.027	0.155
Self-criticism	0.035		0.117	0.515
Emotional expression	−0.001		−0.059	0.037
Wishful thinking	0.019		0.052	0.295
Social support	0.01		0.007	0.187
Cognitive restructuring	0.005		−0.024	0.123
Social withdrawal	0.024		0.032	0.4
Problem avoidance	0		−0.016	0.022
Total effects (Metacognition → mediating variables → Eating disorders)	0.602	<0.001	4.892	5.801

Note: LLCI = lower-limit confidence intervals; ULCI = upper-limit confidence interval.

## Data Availability

The data presented in this study are available on request from the corresponding author. The data are not publicly available because they include confidential information of the participants.

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
