# Peer review of "The Mediating Effect of Coping Strategies and Emotion Regulation in the Relationship between Impulsivity, Metacognition, and Eating Disorders"

_nutrients, 2024, doi:10.3390/nu16121884_

Round 1
Reviewer 1 Report
Comments and Suggestions for Authors
The paper reports a study about the possible relationships between impulsivity, metacognition and emotional support, looking at their connections with eating disorders symptoms. The paper involved a large sample of people, enrolled from the general population. The paper is well-written and clear, but I have a few comments aimed to improve the manuscript.
- Please considered that impulsivity in eating disorders have been evaluated broadly, showing that different diagnoses are linked to different impulsive profile.
- Do you think that the unbalanced ratio between genders might affect the results?
- Is there any specific reason linked to the choice of the Barrat scale for impulsivity, while most studies now use the upps scale?
- this is a cross-sectional study. The use of mediation analyses should be justified due to the critiques raised with this type of approach with non-longitudinal data.
- did you evaluate genders and sexual orientations? these elements could have specific effects on the results and might be considered. Otherwise, please recognize these limits.
- please, include more information about your enrollment strategies.
- please consider to rephrase the "healthy weight" label.
- my main concerns regards the presence of several significant differences between genders that had not been considered as covariate or as elements that might impact the results.
Author Response
We are very grateful for your comments. We have carefully considered and reviewed each of the concerns raised. Below, we provide a detailed description of the amendments we have made to the manuscript. Please note that all changes to the manuscript are highlighted in yellow.
- Please considered that impulsivity in eating disorders have been evaluated broadly, showing that different diagnoses are linked to different impulsive profile.
Thank you for this suggestion. We have now added this issue in the introduction section.
- Do you think that the unbalanced ratio between genders might affect the results?
We thank the reviewer for pointing this out. We acknowledge that the unbalanced gender ratio in our study might influence the results. In our sample, the gender ratio was 22% males to 78% females. For this reason, two mediation models were proposed, one for women and the other for men. However, the results were the same, perhaps due to the absence of differences in the dependent and independent variables. If the reviewer considers that performing a mediation analysis for each gender would significantly improve the quality of the study, the authors are willing to add it. Also, it is worth mentioning that this ratio reflects the demographic characteristics of the population we are studying.
- Is there any specific reason linked to the choice of the Barrat scale for impulsivity, while most studies now use the upps scale?
We acknowledge that the UPPS scale is widely used and provides a comprehensive assessment of impulsivity by distinguishing between various facets such as positive/negative urgency, lack of premeditation, lack of perseverance, and sensation-seeking.
However, both questionnaires have excellent psychometric properties, both in their original and in the Spanish validation. The choice of the BIS scale is due to the subscales, which have been studied in depth in relation to eating behaviour. Furthermore, previous studies in this field have used the BIS to assess impulsive choice (e.g., see Bénard et al., 2019; Goodwin et al., 2023; Shaker et al., 2022)
Bénard, M., Bellisle, F., Kesse-Guyot, E., Julia, C., Andreeva, V. A., Etilé, F., Reach, G., Dechelotte, P., Tavolacci, M. P., Hercberg, S., & Péneau, S. (2019). Impulsivity is associated with food intake, snacking, and eating disorders in a general population. The American Journal of Clinical Nutrition, 109(1), 117–126. https://doi.org/10.1093/ajcn/nqy255
Goodwin, A. L., Butler, G. K. L., & Nikčević, A. V. (2023). Impulsivity dimensions and their associations with disinhibited and actual eating behaviour. Eating Behaviors, 49, 101752. https://doi.org/10.1016/j.eatbeh.2023.101752
Shaker, N. M., Azzam, L. A., Zahran, R. M., & Hashem, R. E. (2022). Frequency of binge eating behavior in patients with borderline personality disorder and its relation to emotional regulation and impulsivity. Eating and Weight Disorders: EWD, 27(7), 2497–2506. https://doi.org/10.1007/s40519-022-01358-x
- This is a cross-sectional study. The use of mediation analyses should be justified due to the critiques raised with this type of approach with non-longitudinal data.
We thank the reviewer for pointing out this limitation. Several studies have successfully used mediation analysis in cross-sectional designs to test theoretical models and understand potential mechanisms (see Winer et al., 2016). While longitudinal data would provide stronger evidence for causal pathways, cross-sectional mediation analysis can still offer valuable insights. We have used bootstrapping techniques to assess the indirect effects, which helps to provide more robust estimates even within the constraints of cross-sectional data.
For these reasons, we have included in the limitations section the impossibility of confirming causality, which limits the robustness of our conclusions about mediating relationships (lines 286-287).
Winer, E. S., Cervone, D., Bryant, J., McKinney, C., Liu, R. T., & Nadorff, M. R. (2016). Distinguishing mediational models and analyses in clinical psychology: Atemporal associations do not imply causation. Journal of Clinical Psychology, 72(9), 947–955. https://doi.org/10.1002/jclp.22298
- Did you evaluate genders and sexual orientations? these elements could have specific effects on the results and might be considered. Otherwise, please recognize these limits.
We thank the reviewer for this comment. We acknowledge that sexual orientation can have specific effects on the results and is important to consider. However, it was not included as a variable in the study because it was outside the stated objectives.
Differences in study variables based on gender are displayed in Table 1.
- Please, include more information about your enrollment strategies.
We thank the reviewer for this comment. We have now expanded the procedure section (see lines 108-113).
- Please consider to rephrase the "healthy weight" label.
Thanks. Done.
- My main concerns regards the presence of several significant differences between genders that had not been considered as covariate or as elements that might impact the results.
We thank the reviewer for underscoring this issue. As previously stated, two mediation models were initially proposed, one for women and the other for men. However, the same results were obtained, possibly due to the absence of significant differences in the dependent and independent variables (metacognition, impulsivity, and eating behaviour). Should the reviewer consider that conducting a mediation analysis for each sex would significantly enhance the quality of the study, the authors are willing to incorporate this.

Reviewer 2 Report
Comments and Suggestions for Authors
This paper presents data from a large sample regarding three hypotheses. Some points for a major revision:
The title seems misleading. It would be better to mention in the title that only female participants were included and therefore rerun statistical analyses only in women participants (as they seem to be the majority in this sample).
The introduction and discussion should be more extended with more references.
The last paragraph of the introduction presents the hypotheses, but they are not supported by relevant references. For example, a recent relevant article with a plethora of references to find in it is the following https://doi.org/10.3390/healthcare12090925
The recruitment method is not clearly presented in the Methods section.
It would be useful to focus only on one type of eating disoders or mention exactly from what participants suffered and the relevant diagnostic criteria followed.
The authors need to explain the tables and figures in the main text.
In the discussion authors can discuss in a critical way the implications for clinical settings and therapy e.g. metacognitive therapy (see and discuss a relevant recent article on what therapists perceive regarding this topic https://doi.org/10.1080/13642537.2023.2278088).
Comments on the Quality of English LanguageModerate English language editing.
Author Response
This paper presents data from a large sample regarding three hypotheses. Some points for a major revision:
We thank the reviewer for the comments about our manuscript. All the suggestions have been taken into account. Below is a point-by-point response to each of the amendments/clarifications requested. Please note that changes to the manuscript are highlighted in yellow.
The title seems misleading. It would be better to mention in the title that only female participants were included and therefore rerun statistical analyses only in women participants (as they seem to be the majority in this sample).
We thank the reviewer for this suggestion. However, the objective of this study was to examine the relationship between the variables in this population, including both males and females. The ratio of men to women represents the gender distribution observed in university studies where recruitment was mostly performed. Notably, the limitations section includes the fact that the sample is composed mainly of women and young students and, therefore, warns against extrapolating the results to other populations (see lines 287-288).
The introduction and discussion should be more extended with more references.
We thank the reviewer for this comment. We have now included more references in the introduction and discussion sections.
The last paragraph of the introduction presents the hypotheses, but they are not supported by relevant references. For example, a recent relevant article with a plethora of references to find in it is the following https://doi.org/10.3390/healthcare12090925
We thank the reviewer for providing valuable references for our manuscript.
The recruitment method is not clearly presented in the Methods section.
Thanks. We have now expanded the procedure section (see lines 108-113).
It would be useful to focus only on one type of eating disorders or mention exactly from what participants suffered and the relevant diagnostic criteria followed.
We thank the reviewer for this comment. The aim of the study is not to analyze the relationship between impulsivity and metacognition in individuals with a specific eating disorder (e.g., anorexia, bulimia). It should be noted that we are dealing with the general population, which does not present any psychological diagnosis. The EDI inventory does not differentiate between these diagnostic categories but rather assesses the severity of the eating behaviour in its different subscales. Therefore, it is difficult to focus on the impact of impulsivity and metacognition on a specific subsample. In this line, diagnostic criteria for an ED in this population were not evaluated.
The authors need to explain the tables and figures in the main text.
The results of the tables are presented in the Results section. The results pertaining to Table 1 are presented in lines 197-203. The correlations are indicated in lines 204-207. Finally, the findings of the mediation analyses are indicated in lines 208-219.
In the discussion authors can discuss in a critical way the implications for clinical settings and therapy e.g. metacognitive therapy (see and discuss a relevant recent article on what therapists perceive regarding this topic https://doi.org/10.1080/13642537.2023.2278088).
We thank the reviewer for pointing this out. We have now added the importance of being aware of the vulnerability, bias, and the difficulties therapists face when dealing with patients with anorexia nervosa (see lines 284-285).

Round 2
Reviewer 1 Report
Comments and Suggestions for Authors
I think the authors have addressed all my concerns.